**Implementation of Yale Interactive terrestrial Biosphere model v1.0**
**into GEOS-Chem v12.0.0: a tool for biosphere-chemistry interactions**
**Yadong Lei[1,2], Xu Yue[3], Hong Liao[3], Cheng Gong[2,4], Lin Zhang[5]**
[1]Climate Change Research Center, Institute of Atmospheric Physics, Chinese
Academy of Sciences, Beijing, 100029, China
[2]University of Chinese Academy of Sciences, Beijing, China
[3]Jiangsu Key Laboratory of Atmospheric Environment Monitoring and Pollution
Control, Collaborative Innovation Center of Atmospheric Environment and
Equipment Technology, School of Environmental Science and Engineering, Nanjing
University of Information Science & Technology (NUIST), Nanjing, 210044, China
[4]State Key Laboratory of Atmospheric Boundary Layer Physics and Atmospheric
Chemistry (LAPC), Institute of Atmospheric Physics, Chinese Academy of Sciences,
Beijing, 100029, China
[5]Laboratory for Climate and Ocean–Atmosphere Studies, Department of Atmospheric
and Oceanic Sciences, School of Physics, Peking University, Beijing, 100871, China
*Correspondence to*: Xu Yue (yuexu@nuist.edu.cn)

**Abstract:** The terrestrial biosphere and atmospheric chemistry interact through multiple feedbacks, but the models of vegetation and chemistry are developed separately. In this study, the Yale Interactive terrestrial Biosphere (YIBs) model, a dynamic vegetation model with biogeochemical processes, is implemented into the Chemical Transport Model GEOS-Chem version 12.0.0. Within the GC-YIBs framework, leaf area index (LAI) and canopy stomatal conductance dynamically predicted by YIBs are used for dry deposition calculation in GEOS-Chem. In turn, the simulated surface ozone ($O_3$) by GEOS-Chem affect plant photosynthesis and biophysics in YIBs. The updated stomatal conductance and LAI improve the simulated $O_3$ dry deposition velocity and its temporal variability for major tree species. For daytime dry deposition velocities, the model-to-observation correlation increases from 0.69 to 0.76 while the normalized mean error (NME) decreases from 30.5% to 26.9% using the GC-YIBs model. For diurnal cycle, the NMEs decrease by 9.1% for Amazon forest, 6.8% for coniferous forest, and 7.9% for deciduous forest using the GC-YIBs model. Furthermore, we quantify $O_3$ vegetation damaging effects and find a global reduction of annual gross primary productivity by 1.5-3.6%, with regional extremes of 10.9–14.1% in the eastern U.S. and eastern China. The online GC-YIBs model provides a useful tool for discerning the complex feedbacks between atmospheric chemistry and the terrestrial biosphere under global change.

**Keywords:** GC-YIBs model, biosphere-chemistry interactions, dry deposition, ozone vegetation damage, evaluations

## 1 Introduction

The terrestrial biosphere interacts with atmospheric chemistry through the exchanges of trace gases, water, and energy (Hungate and Koch, 2015; Green et al., 2017). Emissions from the terrestrial biosphere, such as biogenic volatile organic compounds (BVOCs) and nitrogen oxides ($NO_x$) affect the formation of air pollutants and chemical radicals in the atmosphere (Kleinman, 1994; Li et al., 2019). Globally, the terrestrial biosphere emits ~1100 Tg (1 Tg = $10^{12}$ g) BVOC annually, which is approximately ten times more than the total amount of VOC emitted worldwide from anthropogenic sources including fossil fuel combustion and industrial activities (Carslaw et al., 2010). Meanwhile, the biosphere acts as a major sink through dry deposition of air pollutants, such as surface ozone ($O_3$) and aerosols (Petroff, 2005; Fowler et al., 2009; Park et al., 2014). Dry deposition accounts for ~25% of the total $O_3$ removed from the troposphere (Lelieveld and Dentener, 2000).

In turn, atmospheric chemistry can also affect the terrestrial biosphere (McGrath et al., 2015; Schiferl and Heald, 2018; Yue and Unger, 2018). Surface $O_3$ has a negative impact on plant photosynthesis and crop yields by reducing gas-exchange and inducing phytotoxic damages on plant tissues (Van Dingenen et al., 2009; Wilkinson et al., 2012; Yue and Unger, 2014). Unlike $O_3$, the effect of aerosols on vegetation is dependent on the aerosol concentrations. Moderate increase of aerosols in the atmosphere is beneficial to vegetation (Mahowald, 2011; Schiferl and Heald, 2018). The aerosol-induced enhancement in diffuse light results in more radiation reaching surface from all directions than solely from above. As a result, leaves in the shade or

at the bottom can receive more radiation and are able to assimilate more $CO_2$ through
photosynthesis, leading to an increase of canopy productivity (Mercado et al., 2009;
Yue and Unger, 2018). However, excessive aerosol loadings reduce canopy
productivity because the total radiation is largely weakened (Alton, 2008; Yue and
Unger, 2017).

Models are essential tools to understand and quantify the interactions between the
terrestrial biosphere and atmospheric chemistry at the global and/or regional scales.
Many studies have performed multiple global simulations with
climate-chemistry-biosphere models to quantify the effects of air pollutants on the
terrestrial biosphere (Mercado et al., 2009; Yue and Unger, 2015; Oliver et al., 2018;
Schiferl and Heald, 2018). In contrast, very few studies have quantified the
$O_3$-induced biogeochemical and meteorological feedbacks to air pollution
concentrations (Sadiq et al., 2017; Zhou et al., 2018). Although considerable efforts
have been made, uncertainties in biosphere-chemistry interactions remain large
because their two-way coupling is not adequately represented in current generation of
terrestrial biosphere models or global chemistry models. Global terrestrial biosphere
models usually use prescribed $O_3$ and aerosol concentrations (Sitch et al., 2007;
Mercado et al., 2009; Lombardozzi et al., 2012), and global chemistry models often
apply fixed offline vegetation variables (Lamarque et al., 2013). For example,
stomatal conductance, which plays a crucial role in regulating water cycle and altering
pollution deposition, responds dynamically to vegetation biophysics and
environmental stressors at various spatiotemporal scales (Hetherington and Woodward,
2003; Franks et al., 2017). However, these processes are either missing or lack of
temporal variations in most current chemical transport models (Verbeke et al., 2015).
The fully two-way coupling between biosphere and chemistry is necessary to better
quantify the responses of ecosystems and pollution to global changes.

In this study, we develop the GC-YIBs model by implementing the Yale Interactive
terrestrial Biosphere (YIBs) model version 1.0 (Yue and Unger, 2015) into the
chemical    transport    model    (CTM)    GEOS-Chem    version    12.0.0
(http://wiki.seas.harvard.edu/   geos-chem/index.php/GEOS-Chem_12#12.0.0).    The
GEOS-Chem (short as GC thereafter) model has been widely used in episode
prediction (Cui et al., 2016), source attribution (D'Andrea et al., 2016; Dunker et al.,
2017; Ni et al., 2018; Lu et al., 2019), future pollution projection (Yue et al., 2015;
Ramnarine et al., 2019), health risk assessment (Xie et al., 2019), and so on. The
standard GC model uses prescribed vegetation parameters and as a result cannot
depict the changes in chemical components due to biosphere-pollution interactions.
The updated GC-YIBs model links atmospheric chemistry with biosphere in a
two-way coupling such that changes in chemical components or vegetation will
simultaneously feed back to influence the other systems. Here, we evaluate the
dynamically simulated dry deposition and leaf area index (LAI) from GC-YIBs and
examine the consequent impacts on surface $O_3$. We also quantify the detrimental
effects of $O_3$ on gross primary productivity (GPP) using instant pollution
concentrations from the chemical module. For the first step, we focus on the coupling
between $O_3$ and vegetation. The interactions between aerosols and vegetation will be
developed and evaluated in the future. The next section describes the GC-YIBs model
and the evaluation data. Section 3 compares simulated $O_3$ from GC-YIBs with that
from the original GC models and explores the causes of differences. Section 4
quantifies $O_3$ damaging effects to global GPP using the GC-YIBs model. The last
section summarizes progresses and discusses the next-step tasks to optimize the
GC-YIBs model.

**2 Methods and data**
**2.1 Descriptions of the YIBs model**
YIBs is a terrestrial vegetation model designed to simulate land carbon cycle with
dynamical prediction of LAI and tree height (Yue and Unger, 2015). The YIBs model
applies Farquhar et al. (1980) scheme to calculate leaf level photosynthesis, which is
further upscaled to canopy level by the separation of sunlit and shaded leaves (Spitters,
1986). The canopy is divided into an adaptive number of layers (typically 2-16) for
light stratification. Sunlight is attenuated and becomes more diffusive when
penetrating the canopy. The sunlit leaves can receive both direct and diffuse radiation,
while the shading leaves receive only diffuse radiation. The leaf-level photosynthesis,
calculated as the sum of sunlit and shading leaves, is then integrated over all canopy
layers to derive the GPP of ecosystems.

The model considers 9 plant functional types (PFTs), including evergreen needleleaf
forest, deciduous broadleaf forest, evergreen broadleaf forest, shrubland, tundra,
$C_3/C_4$ grasses, and $C_3/C_4$ crops. The satellite-based land types and cover fraction are
aggregated into these 9 PFTs and used as input (Fig. S1). The initial soil carbon pool
and tree height used in YIBs are from the 140 years spin-up processes (Yue and Unger,
2015). The YIBs is driven with hourly 2-D meteorology and 3-D soil variables (6
layers) from the Modern-Era Retrospective analysis for Research and Applications,
version 2 (MERRA2).

The YIBs uses the model of Ball and Berry (Baldocchi et al., 1987) to compute leaf
stomatal conductance:
$$g_s = \frac{1}{r_s} = m \frac{A_{net}}{c_s} RH + b \tag{1}$$
where $r_s$ is the leaf stomatal resistance ($s\ m^{-1}$); $m$ is the empirical slope of the
Ball-Berry stomatal conductance equation and is affected by water stress; $c_s$ is the
$CO_2$ concentration at the leaf surface ($\mu mol\ m^{-3}$); $RH$ is the relative humidity of
atmosphere; $b$ ($m\ s^{-1}$) represents the minimum leaf stomatal conductance when net
leaf photosynthesis ($A_{net}, \mu mol\ m^{-2}\ s^{-1}$) is 0. For different PFTs, appropriate
photosynthetic parameters are derived from the Community Land Model (CLM)
(Bonan et al., 2011).

The net leaf photosynthesis for $C_3$ and $C_4$ plants is computed based on
well-established Michaelis–Menten enzyme-kinetics scheme (Farquhar et al., 1980;
Voncaemmerer and Farquhar, 1981):

$$A_{net} = min(J_c, J_e, J_s) - R_d \qquad (2)$$

Where $J_c$, $J_e$ and $J_s$ represent the Rubiso-limited photosynthesis, the RuBP-limited
photosynthesis, and the product-limited photosynthesis, respectively. $R_d$ is the rate of
dark respiration. They are all parameterized as functions of the maximum
carboxylation capacity (Collatz et al., 1991) and meteorological variables (e.g.,
temperature, radiation, and $CO_2$ concentrations).

The YIBs model applies the LAI and carbon allocation schemes from the TRIFFID
model (Cox, 2001; Clark et al., 2011). On the daily scale, canopy LAI is calculated as
follows:

$$LAI = f \times LAI_{max} \qquad (3)$$

where $f$ represents phenological factor controlled by meteorological variables (e.g.,
temperature, water availability, and photoperiod); $LAI_{max}$ represents the available
maximum LAI related to tree height, which is dependent on the vegetation carbon
content ($C_{veg}$). The $C_{veg}$ is calculated as follows:

$$C_{veg} = C_l + C_r + C_w \qquad (4)$$

where $C_l$, $C_r$ and $C_w$ represent leaf, root, and stem carbon contents, respectively.
And all carbon components are parameterized as the function of $LAI_{max}$:
$$\begin{cases} C_l = \alpha \times LAI \\ C_r = \alpha \times LAI_{max} \\ C_r = \beta \times LAI_{max}^{\gamma} \end{cases} \qquad (5)$$
where $\alpha$ represents the specific leaf carbon density; $\beta$ and $\gamma$ represent allometric
parameters. The vegetation carbon content $C_{veg}$ is updated every 10 days:

$$\frac{dC_{veg}}{dt} = (1 - \tau) \times NPP - \varphi \tag{6}$$

where $\tau$ and $\varphi$ represent partitioning parameter and litter fall rate, respectively, and their calculation methods have been documented in Yue and Unger (2015). Net primary productivity (NPP) is calculated as the residue of subtracting autotrophic respiration ($R_a$) from GPP:

$$NPP = GPP - R_a \tag{7}$$

In addition, the YIBs model implements the scheme for $O_3$ damage on vegetation proposed by Sitch et al. (2007). The scheme directly modifies photosynthesis using a semi-mechanistic parameterization, which in turn affects stomatal conductance. The $O_3$ damage factor is considered as the function of stomatal $O_3$ flux:

$$F = \begin{cases} -a(F_{O_3} - T_{O_3}), & F_{O_3} > T_{O_3} \\ 0, & F_{O_3} \leq T_{O_3} \end{cases} \tag{8}$$

Where $a$ represents the damaging sensitivity and $T_{O_3}$ represents the $O_3$ flux threshold ($\mu mol\ m^{-2}\ s^{-1}$). For a specific PFT, the values of coefficient $a$ vary from low to high to represent a range of uncertainties for ozone vegetation damaging (Table S1). $T_{O_3}$ is a critical threshold for $O_3$ damage and varies with PFTs. The $F$ becomes negative only if $F_{O_3}$ is higher than $T_{O_3}$. Stomatal $O_3$ flux $F_{O_3}$ ($\mu mol\ m^{-2}\ s^{-1}$) is calculated as follows:

$$F_{O_3} = \frac{[O_3]}{r_a + r_b + k \cdot r_s} \tag{9}$$

where $[O_3]$ represents $O_3$ concentrations at top of the canopy ($\mu mol\ m^{-3}$); $r_a$ is aerodynamic resistance ($s\ m^{-1}$) and $r_b$ is boundary layer resistance ($s\ m^{-1}$); $r_s$ represents stomatal resistance ($s\ m^{-1}$). The Sitch et al. (2007) scheme within the YIBs

framework has been well evaluated against hundreds of observations globally (Yue
and Unger, 2018) and regionally (Yue et al., 2016; Yue et al., 2017).

**2.2 Descriptions of the GEOS-Chem model**
GC is a global 3-D model of atmospheric compositions with fully coupled
$O_3$-$NO_x$-hydrocarbon-aerosol chemical mechanisms (Gantt et al., 2015; Lee et al.,
2017; Ni et al., 2018). In this study, we use GC version 12.0.0 driven by assimilated
meteorology from MERRA2 with a horizontal resolution of 4° latitude by 5°
longitude and 47 vertical layers from surface to 0.01 hPa.

In GC, terrestrial vegetation modulates tropospheric $O_3$ mainly through LAI and
canopy stomatal conductance, which affect both the sources and sinks of tropospheric
$O_3$ through changes in BVOC emissions, soil $NO_x$ emissions, and dry deposition
(Zhou et al., 2018). BVOC emissions are calculated based on a baseline emission
factor parameterized as the function of light, temperature, leaf age, soil moisture, LAI,
and $CO_2$ inhibition within the Model of Emissions of Gasses and Aerosols from
Nature (MEGAN v2.1) (Guenther et al., 2006). Soil $NO_x$ emission is computed based
on the scheme of Hudman et al. (2012) and further modulated by a reduction factor to
account for within-canopy $NO_x$ deposition (Rogers and Whitman, 1991). The dry
deposition velocity ($V_d$, $m\,s^{-1}$) for $O_3$ is computed based on a resistance-in-series
model within GC:
$$V_d = \frac{1}{R_a + R_b + R_c} \qquad (10)$$
where $R_a$ ($m\ s^{-1}$) is the aerodynamic resistance representing the ability of the
airflow to bring gases or particles close to the surface and is dependent mainly on the
atmospheric turbulence structure and the height considered. $R_b$ ($m\ s^{-1}$) is the
boundary resistance driven by the characteristics of surface (surface roughness) and
gas/particle (molecular diffusivity). $R_a$ and $R_b$ are calculated from the global
climate models (GCM) meteorological variables (Jacob et al., 1992). The surface
resistance $R_c$ is determined by the affinity of surface for the chemical compound. For
$O_3$ over vegetated regions, $V_d$ is mainly driven by $R_c$ ($m\ s^{-1}$) during daytime
because the effects of $R_a$ and $R_b$ are generally small. Surface resistances $R_c$ are
computed using the Wesely (1989) canopy model with some improvements, including
explicit dependence of canopy stomatal resistances on LAI (Gao and Wesely, 1995)
and direct/diffuse PAR within the canopy (Baldocchi et al., 1987):
$$\frac{1}{R_c} = \frac{1}{R_s + R_m} + \frac{1}{R_{lu}} + \frac{1}{R_{cl}} + \frac{1}{R_g} \tag{11}$$

where $R_s$ is the stomatal resistance ($s\ m^{-1}$), $R_m$ is the leaf mesophyll resistance
($R_m = 0\ s\ m^{-1}$ for $O_3$), $R_{lu}$ is the upper canopy or leaf cuticle resistance, $R_{cl}$ is the
lower canopy resistance ($s\ m^{-1}$). $R_s$ is calculated based on minimum stomatal
resistance ($r_s$, $s\ m^{-1}$), solar radiation ($G$, $W\ m^{-2}$), surface air temperature ($T_s$, °C),
and the molecular diffusivities ($D_{H_2O}$ and $D_x$) for a specific gas $x$:
$$R_s = r_s \left\{ 1 + \frac{1}{[200(G+0.1)]^2} \right\} \left\{ \frac{400}{T_s(40-T_s)} \right\} \frac{D_{H_2O}}{D_x} \tag{12}$$

In GC, the above parameters related to $R_c$ have prescribed values for 11 deposition
land types, including snow/ice, deciduous forest, coniferous forest, agricultural land,
shrub/grassland, amazon forest, tundra, desert, wetland, urban and water (Wesely,
1989; Jacob et al., 1992).

Although the stomatal conductance scheme of Wesely (1989) has been widely used in
chemical transport and climate models, considerable limits still exist because this
scheme does not consider the response of stomatal conductance to phenology, $CO_2$
concentrations, and soil water availability (Rydsaa et al., 2016; Lin et al., 2017).
Previous studies have well evaluated the dry deposition scheme used in the
GEOS-Chem model against observations globally and regionally (Hardacre et al.,
2015; Silva and Heald, 2018; Lin et al., 2019; Wong et al., 2019). They found that
GEOS-Chem can generally capture the diurnal and seasonal cycles except for the
amplitude of $O_3$ dry deposition velocity (Silva and Heald, 2018).

**2.3 Implementation of YIBs into GEOS-Chem (GC-YIBs)**
In this study, GC model time steps are set to 30 min for transport and convection and
60 min for emissions and chemistry. In the online GC-YIBs configuration, GC
provides the hourly meteorology, aerodynamic resistance, boundary layer resistance,
and surface [$O_3$] to YIBs. Without YIBs implementation, the GC model computes $O_3$
dry deposition velocity using prescribed LAI and parameterized canopy stomatal
resistance ($R_s$), and as a result ignore feedbacks from ecosystems (details in 2.2).
With YIBs embedded, daily LAI and hourly stomatal conductance are dynamically
predicted for the dry deposition scheme within the GC model. The online-simulated
surface [$O_3$] affects carbon assimilation and canopy stomatal conductance, in turn, the
online-simulated vegetation variables such as LAI and stomatal conductance affect
both the sources and sinks of $O_3$ by altering precursor emissions and dry deposition at
the 1-hour integration time step. The above processes are summarized in Fig. 1.

To retain the corresponding relationship between vegetation parameters and land
cover map in the GC-YIBs model, we replace the Olson 2001 land cover map in GC
with satellite-retrieved land cover dataset used by YIBs (Defries et al., 2000;
Hanninen and Kramer, 2007). The conversion relationships between YIBs land types
and GC deposition land types are summarized in Table S2. The global spatial pattern
of deposition land types converted from YIBs land types is shown in Fig. S2. The
Olson 2001 land cover map used in GC version 12.0.0 has a native resolution of
0.25°×0.25° and 74 land types (Olson et al., 2001). Each of the Olson land types is
associated with a corresponding deposition land type with prescribed parameters.
There are 74 Olson land types but only 11 deposition land types, suggesting that many
of the Olson land types share the same deposition parameters. At specific grids (4°×5°
or 2°×2.5°), dry deposition velocity is calculated as the weighted sum of native
resolution (0.25°×0.25°). Replacing of Olson with YIBs land types induces global
mean difference of -0.59 ppbv on surface $[O_3]$ (Fig. S3). Large discrepancies are
found in Africa and southern Amazon, where the local $[O_3]$ decreases by more than 2
ppbv with the new land types. However, limited differences are shown in mid-high
latitudes of Northern Hemisphere (NH, Fig. S3).

**2.4 Model simulations**

We conduct six simulations to evaluate the performance of GC-YIBs and to quantify global $O_3$ damage to vegetation (Table 1): (i) Offline, a control run using the offline GC-YIBs model. The YIBs module shares the same meteorological forcing as the GC module and predicts both GPP and LAI. However, predicted vegetation variables are not fed into GC, which is instead driven by prescribed LAI from Moderate Resolution Imaging Spectroradiometer (MODIS) product and parameterized canopy stomatal conductance proposed by Gao and Wesely (1995). (ii) Online_LAI, a sensitive run using online GC-YIBs with dynamically predicted daily LAI from YIBs but prescribed stomatal conductance. (iii) Online_GS, another sensitive run using YIBs predicted stomatal conductance but prescribed MODIS LAI. (iv) Online_ALL, in which both YIBs predicted LAI and stomatal conductance are used for GC. (v) Online_ALL_HS, the same as Online_ALL except that predicted surface $O_3$ damages plant photosynthesis with high sensitivities. (vi) Online_ALL_LS, the same as Online_ALL_HS but with low $O_3$ damaging sensitivities. Each simulation is run from 2006 to 2012 with the first 4 years for spin-up, and results from 2010 to 2012 are used to evaluate the online GC-YIBs model. The differences between Online_ALL and Online_GS (Online_LAI) represent the effects of coupled LAI (stomatal conductance) on simulated $[O_3]$. Differences between Offline and Online_ALL then represent joint effects of coupled LAI and stomatal conductance. The last three runs are used to quantify the global $O_3$ damage on ecosystem productivity.

**2.5 Evaluation data**

We use observed LAI data for 2010–2012 from the MODIS product. Benchmark GPP product of 2010–2012 is estimated by upscaling ground-based FLUXNET eddy covariance data using a model tree ensemble approach, a type of machine learning technique (Jung et al., 2009). Although these products may have certain biases, they have been widely used to evaluate land surface models because direct observations of GPP and LAI are not available on the global scale (Yue and Unger, 2015; Slevin et al., 2017; Swart et al., 2019). Measurements of surface [$O_3$] over North America and Europe are provided by the Global Gridded Surface Ozone Dataset (Sofen et al., 2016), and those over China are interpolated from data at ~1500 sites operated by China's Ministry of Ecology and Environment (http://english.mee.gov.cn). We perform literature research to collect data of dry deposition velocity from 8 deciduous forest, 2 amazon forest, and 9 coniferous forest sites (Table 2).

**3 Results**

**3.1 Evaluation of offline GC-YIBs model**

With Offline simulation, the simulated GPP and LAI are compared with observed LAI and benchmark GPP for the period of 2010-2012 (Fig. 2). Observed LAI and benchmark GPP both show high values in the tropics and medium values in the northern mid-high latitudes. Compared to observations, the GC-YIBs model forced with MERRA2 meteorology depicts similar spatial distributions, with spatial correlation coefficients of 0.83 ($p$ <0.01) for GPP and 0.86 ($p$ <0.01) for LAI.

Although the model overestimates LAI in the tropics and northern high latitudes by
1-2 $m^2$ $m^{-2}$, the simulated global area-weighted LAI (1.42 $m^2$ $m^{-2}$) is close to
observations (1.33 $m^2$ $m^{-2}$) with a normalized mean bias (NMB) of 6.7%. Similar to
LAI, the global NMB for GPP is only 7.1%, though there are substantial regional
biases especially in Amazon and central Africa. Such differences are in part attributed
to the underestimation of GPP for tropical rainforest in the benchmark product,
because the recent simulations at eight rainforest sites with YIBs model reproduced
ground-based observations well (Yue and Unger, 2018).

We then evaluate simulated annual mean surface [$O_3$] during 2010-2012 based on
Offline simulation (Fig. 3). The simulated high values are mainly located in the
mid-latitudes of NH (Fig. 3a). Compared to observations, simulations show
reasonable spatial distribution with a correlation coefficient of 0.63 ($p$ <0.01).
Although offline GC-YIBs model overestimates annual [$O_3$] in southern China and
predicts lower values in western Europe and western U.S., the simulated
area-weighted surface [$O_3$] (45.4 ppbv) is only 6% higher than observations (42.8
ppbv). Predicted summertime surface [$O_3$] instead shows positive biases in eastern
U.S. and Europe (Fig. S4), consistent with previous evaluations using the GC model
(Travis et al., 2016; Schiferl and Heald, 2018; Yue and Unger, 2018).

**3.2 Changes of surface $O_3$ in online GC-YIBs model**
Surface $O_3$ is changed by the coupling of LAI and stomatal conductance (Fig. 4).
Global [$O_3$] shows similar patterns between Offline (Fig. 3a) and Online_ALL (Fig.
4a) simulations. However, the online GC-YIBs predicts higher [$O_3$] by 0.5-2 ppbv in
the mid-high latitudes of NH, leading to an average enhancement of [$O_3$] by 0.22
ppbv compared to Offline simulations (Fig. 4b). Regionally, some negative changes of
1-2 ppbv can be found at the tropical regions. With sensitivity experiments
Online_LAI and Online_GS (Table. 1), we separate the contributions of LAI and
stomatal conductance changes to $\Delta$[$O_3$]. It is found that $\Delta$[$O_3$] between Online_ALL
and Online_LAI (Fig. 4c) resembles the total $\Delta$[$O_3$] pattern (Fig. 4b), suggesting that
changes in stomatal conductance play the dominant role in regulating surface [$O_3$]. As
a comparison, $\Delta$[$O_3$] values between Online_ALL and Online_GS show limited
changes globally (by 0.05 ppbv) and moderate changes in tropical regions (Fig. 4d),
mainly because the LAI predicted by YIBs is close to MODIS LAI used in GC (Fig.
2). It is noticed that the average $\Delta$[$O_3$] in Fig. 4b is not equal to the sum of Fig. 4c and
Fig. 4d, because of the non-linear effects.

We further explore the possible causes of differences in simulated [$O_3$] between online
and offline GC-YIBs models. Fig. 5 shows simulated annual $O_3$ dry deposition
velocity from online GC-YIBs model and its changes in different sensitivity
experiments. The global average velocity is 0.25 cm s$^{-1}$ with regional maximum of
0.5-0.7 cm s$^{-1}$ in tropical rainforest (Fig. 5a), especially over Amazon and central
Africa where high ecosystem productivity is observed (Fig. 2). With implementation
of YIBs into GC, simulated dry deposition velocity increases over tropical regions but
decreases in mid-high latitudes of NH (Fig. 5b). Larger dry deposition results in lower
$[O_3]$ in the tropics, while smaller dry deposition increases $[O_3]$ in boreal regions. Such
spatial patterns are broadly consistent with $\Delta[O_3]$ in online GC-YIBs (Fig. 4b). In a
comparison, updated LAI induces limited changes in the isoprene and $NO_x$ emissions
(Fig. S5), suggesting that changes of dry deposition velocity are the dominant drivers
of $O_3$ changes. Both the updated LAI and stomatal conductance influence dry
deposition. Sensitivity experiments further show that changes in dry deposition are
mainly driven by coupled canopy stomatal conductance (Fig. 5c) instead of LAI (Fig.
5d), though the latter contributes to the enhanced dry deposition in the tropics.

The original GC dry deposition scheme applies fixed parameters for stomatal
conductance of a specific land type. The updated GC-YIBs model instead calculates
stomatal conductance as a function of photosynthesis and environmental forcings
(Equation 1). As a result, predicted dry deposition exhibits discrepancies among
biomes. With Offline and Online_ALL simulations, we further evaluate the
performance of online GC-YIBs in simulating $O_3$ dry deposition velocity for specific
deposition land types (Fig. 6). For agricultural land and shrub/grassland, the simulated
$O_3$ dry deposition velocity for online GC-YIBs model is close to GC model with
NMBs of 3%, -2% and correlation coefficients of 0.96, 0.97, respectively. However,
the simulated dry deposition velocity in online GC-YIBs is lower than GC by 18% for
deciduous forest and 14% coniferous forest, but larger by 17% for Amazon forest.
Such changes match the spatial pattern of dry deposition shown in Fig. 5b.

Since the changes of $O_3$ dry deposition velocity are mainly found in deciduous forest,

coniferous forest, and amazon forest, we collect 27 samples across these three biomes

to evaluate the online GC-YIBs model (Table. 2). For the 11 samples at deciduous

forest, the normalized mean error (NME) decreases from 29% in GC model to 24% in

GC-YIBs with lower relative errors in 8 sites (Fig. 7). Predictions with the GC-YIBs

also show large improvements over coniferous forest, where 8 out of 14 samples

showing lower (decreases from 27% in GC to 25% in GC-YIBs) errors. For amazon

forest, the GC-YIBs model significantly improves the prediction at one site (117.9°E,

4.9°N) where the original error of -0.17 cm s$^{-1}$ is limited to only 0.03 cm s$^{-1}$. However,

the new model does not improve the prediction at the other amazon forest site. Overall,

the simulated daytime $O_3$ dry deposition velocities in online GC-YIBs model are

closer to observations than those in GC model with smaller NME (26.9% vs. 30.5%),

root-mean-square errors (RMSE, 0.2 vs. 0.23) and higher correlation coefficients

(0.76 vs. 0.69). Such improvements consolidate our strategies in updating GC model

to the fully coupled GC-YIBs model.

We collect long-term measurements from 4 sites across northern America and western

Europe to evaluate the model performance in simulating seasonal cycle of $O_3$ dry

deposition velocity (Fig. 8). The GC model well captures the seasonal cycles of $O_3$

dry deposition velocity in all sites with the correlation coefficients of 0.95 in Harvard,

0.8 in Hyytiala, 0.68 in Ulborg, and 0.71 in Auchencorth. However, the magnitude of

$O_3$ dry deposition velocity is overestimated in Harvard and Hyytiala sites (NME of 60%
and 42%, respectively) but underestimated in Ulborg and Auchencorth sites (NME of
48.7% and 58.9%, respectively) at growing seasons. Compared to the GC model,
simulated $O_3$ dry deposition velocity with the GC-YIBs model shows large
improvements over Harvard (Hyytiala) sites, where the model-to-observation NME
decreases from 60% (42%) to 32% (28%).

Additionally, we investigate the diurnal cycle of $O_3$ dry deposition velocity at 15 sites
(Fig. S6). Observed $O_3$ dry deposition velocities show single diurnal peak with the
maximum from local 8 a.m. to 4 p.m. (Fig. 9). Compared to observations, the GC
model has good performance in simulating the diurnal cycle with correlation
coefficients of 0.94 for Amazon forest, 0.96 for coniferous forest, and 0.95 for
deciduous forest. The GC model underestimates daytime $O_3$ dry deposition velocity at
Amazon forest (NME of 29.8%) but overestimates it at coniferous and deciduous
forests (NME of 21.9% and 22.9%, respectively). Compared to the GC model, the
simulated daytime $O_3$ dry deposition velocities using the GC-YIBs model are closer to
observations in all three biomes. The NMEs decrease by 9.1% for Amazon forest, 6.8%
for coniferous forest, and 7.9% for deciduous forest.

**3.3 Assessment of global $O_3$ damages to vegetation**
An important feature of GC-YIBs is the inclusion of online vegetation damages by
surface $O_3$. Here, we quantify the global $O_3$ damages to GPP and LAI by conducting
Online_ALL_HS and Online_ALL_LS simulations (Fig. 10). Due to $O_3$ damaging,
annual GPP declines from -1.5% (low sensitivity) to -3.6% (high sensitivity) on the
global scale. Regionally, $O_3$ decreases GPP as high as 10.9% in the eastern U.S. and
up to 14.1% in eastern China at the high sensitivity (Figs. 10a, b). Such strong
damages are related to (i) high ambient $[O_3]$ due to anthropogenic emissions and (ii)
large stomatal conductance due to active ecosystem productivity in monsoon areas.
The $O_3$ effects are moderate in tropical areas, where stomatal conductance is also high
while $[O_3]$ is very low (Fig. 4a) due to limited anthropogenic emissions. Furthermore,
$O_3$-induced GPP reductions are also small in western U.S. and western Asia. Although
$[O_3]$ is high over these semi-arid regions (Fig. 4a), the drought stress decreases
stomatal conductance and consequently constrains the $O_3$ uptake. The damages to LAI
(Figs. 10c, d) generally follow the pattern of GPP reductions (Figs. 10a, b) but with
lower magnitude. The reductions of GPP are slightly higher than our previous
estimates using prescribed LAI and/or surface $[O_3]$ in the simulations (Yue and Unger,
2014, 2015), likely because GC-YIBs considers $O_3$-vegetation interactions. The
feedback of such interaction to both chemistry and biosphere will be explored in
future studies.

**4 Conclusions and discussion**
The terrestrial biosphere and atmospheric chemistry interact through a series of
feedbacks (Green et al., 2017). Among biosphere-chemistry interactions, dry
deposition plays a key role in the exchange of compounds and acts as an important
sink for several air pollutants (Verbeke et al., 2015). However, dry deposition is
simply parameterized in most of current CTMs (Hardacre et al., 2015). For all
chemical species considered in GC model, stomatal resistance $R_c$ is simply
calculated as the function of minimum stomatal resistance and meteorological
forcings. Such parameterization not only induces biases, but also ignores the
feedbacks from biosphere-chemistry interactions. For example, recent studies
revealed that $O_3$-induced damages to vegetation could reduce stomatal conductance
and in turn alter ambient $O_3$ level (Sadiq et al., 2017; Zhou et al., 2018). In this study,
we implement YIBs into the GC model with fully interactive surface $O_3$ and the
terrestrial biosphere. The dynamically predicted LAI and stomatal conductance from
YIBs are instantly provided to GC, meanwhile the prognostic $O_3$ simulated by GC is
simultaneously affecting vegetation biophysics in YIBs. With these updates, simulated
$O_3$ dry deposition velocities and its temporal variability (seasonal and diurnal cycles)
in GC-YIBs are closer to observations than those in original GC model.

An earlier study updated dry deposition scheme in the Community Earth System
Model (CESM) by implementing the leaf and stomatal resistances (Val Martin et al.,
2014). Compared to that work, the magnitudes of $\Delta[O_3]$ in our simulations are smaller
in northern America, eastern Europe, and southern China. This might be because the
original dry deposition scheme in the GC model (see validation in Fig. 7) is better
than that in CESM, leaving limited potentials for improvements. In GC, the leaf
cuticular resistance ($R_{lu}$) is dependent on LAI (Gao and Wesely, 1995), while the
original calculation of $R_{lu}$ in CESM does not include LAI (Wesely, 1989). In
addition, differences in the canopy schemes for stomatal conductance between YIBs
and Community Land Model (CLM) may cause different responses in dry deposition,
which is changed by -0.12 to 0.16 cm s$^{-1}$ in GC-YIBs but much larger by -0.15 to 0.25
cm s$^{-1}$ in CESM (Val Martin et al., 2014). Moreover, the GC-YIBs is driven with
prescribed reanalysis while CESM dynamically predicts climatic variables.
Perturbations of meteorology in response to terrestrial properties may further magnify
the variations in atmospheric components in CESM.

Although we implement YIBs into GC with fully interactive surface $O_3$ and the
terrestrial biosphere, it should be noted that considerable limits still exist and further
developments are required for GC-YIBs. (1) Atmospheric nitrogen alters plant growth
and further influences both the sources and sinks of surface $O_3$ through surface–
atmosphere exchange processes (Zhao et al., 2017). However, the YIBs model
currently utilizes a fixed nitrogen level and does not include an interactive nitrogen
cycle, which may induce uncertainties in simulating carbon fluxes. (2) Validity of
$\Delta[O_3]$, especially those at high latitudes in NH, cannot be directly evaluated due to a
lack of measurements. Although changes of dry deposition show improvements in
GC-YIBs, the ultimate effects on surface $[O_3]$ remain unclear within the original GC
framework. (3) $[O_3]$ at the lowest model level is used as an approximation of canopy
$[O_3]$. The current model does not include a sub-grid parameterization of pollution
transport within canopy, leading to biases in estimating $O_3$ vegetation damage and the
consequent feedback. However, development of such parameterization is limited by
the availability of simultaneous measurements of microclimate and air pollutants. (4)
The current GC-YIBs is limited to a low resolution due to slow computational speed
and high computational costs for long-term integrations. The GC model, even at the
$2° \times 2.5°$ resolution, takes days to simulate 1 model year due to comprehensive
parameterizations of physical and chemical processes. Such low speed constrains
long-term spin up required by dynamical vegetation models. The low resolution will
affect local emissions (e.g., $NO_x$ and VOC) and transport, leading to changes in
surface $[O_3]$ in GEOS-Chem. The comparison results of 2007 show that low
resolution of $4° \times 5°$ induces a global mean bias of -0.24 ppbv on surface $[O_3]$
compared to the relatively high resolution at $2° \times 2.5°$ (Fig. S7). Compared with
surface $[O_3]$, low resolution causes limited differences in vegetation variables (e.g.,
GPP and LAI, not shown).

Despite these deficits, the development of GC-YIBs provides a unique tool for
studying biosphere-chemistry interactions. In the future, we will extend our
applications in: (1) Air pollution impacts on biosphere, including both $O_3$ and aerosol
effects. The GC-YIBs model can predict atmospheric aerosols, which affect both
direct and diffuse radiation through the Rapid Radiative Transfer Model for GCMs
(RRTMG) in the GC module (Schiferl and Heald, 2018). The diffuse fertilization
effects in the YIBs model have been fully evaluated (Yue and Unger, 2018), and as a
result we can quantify the impacts of aerosols on terrestrial ecosystems. (2) Multiple
schemes for BVOC emissions. The YIBs model incorporates both MEGAN (Guenther

et al., 2006) and photosynthesis-dependent (Unger, 2013) isoprene emission schemes (Yue and Unger, 2015). The two schemes within the GC-YIBs framework can be used and compared for simulations of BVOC and consequent air pollution (e.g., $O_3$, secondary organic aerosols). (3) Biosphere-chemistry feedbacks to air pollution. The effects of air pollution on the biosphere include changes in stomatal conductance, LAI, and BVOC emissions, which in turn modify the sources and sinks of atmospheric components. Only a few studies have quantified these feedbacks for $O_3$-vegetation interactions (Sadiq et al., 2017; Zhou et al., 2018). We can explore the full biosphere-chemistry coupling for both $O_3$ and aerosols using the GC-YIBs model in the future.

**Code availability**

The YIBs model was developed by Xu Yue and Nadine Unger with code sharing at https://github.com/YIBS01/YIBS_site. The GEOS-Chem model was developed by the Atmospheric Chemistry Modeling Group at Harvard University led by Prof. Daniel Jacob and improved by a global community of atmospheric chemists. The source code for the GEOS-Chem model is publicly available at http://acmg.seas.harvard.edu/geos/. The source codes for the GC-YIBs model is archived at https://doi.org/10.5281/zenodo.3659346.

*Author contributions.* Xu Yue conceived the study. Yadong Lei and Xu Yue were responsible for model coupling, simulations, results analysis and paper writing. All

co-authors improved and prepared the manuscript.

*Competing interests.* The authors declare that they have no conflict of interest.

*Acknowledgements.* This work is supported by the National Key Research and
Development Program of China (grant no. SQ2019YFA060013-02) and National
Natural Science Foundation of China (grant no. 41975155).

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

**Table 1** Summary of simulations using the GC-YIBs model.

| Name | Scheme | Ozone effects |
|---|---|---|
| Offline | Monthly prescribed MODIS LAI | No |
| | Original dry deposition scheme | |
| Online_LAI | Daily dynamically predicted LAI | No |
| | Original dry deposition scheme | |
| Online_GS | Monthly prescribed MODIS LAI | No |
| | Hourly predicted stomatal conductance | |
| Online_ALL | Daily dynamically predicted LAI | No |
| | Hourly predicted stomatal conductance | |
| Online_ALL_HS | Daily dynamically predicted LAI | |
| | Hourly predicted stomatal conductance | High |
| | Hourly predicted [$O_3$] by GC model | |
| Online_ALL_LS | Daily dynamically predicted LAI | |
| | Hourly predicted stomatal conductance | Low |
| | Hourly predicted [$O_3$] by GC model | |


**Table 2** List of measurement sites used for dry deposition evaluation.

| Land type | Longitude | Latitude | Season | Daytime $V_d$ (cm s$^{-1}$) | References |
|---|---|---|---|---|---|
| Deciduous forest | 80.9°W | 44.3°N | summer | 0.92 | Padro et al. (1991) |
| | | | winter | 0.28 | |
| | 72.2°W | 42.7°N | summer | 0.61 | Munger et al. (1996) |
| | | | winter | 0.28 | |
| | 75.2°W | 43.6°N | summer | 0.82 | Finkelstein et al. (2000) |
| | 78.8°W | 41.6°N | summer | 0.83 | |
| | 99.7°E | 18.3°N | spring | 0.38 | Matsuda et al. (2005) |
| | | | summer | 0.65 | |
| | 0.84°W | 51.17°N | Jul-Aug | 0.85 | Fowler et al. (2009) |
| | 0.7°W | 44.2°N | Jun | 0.62 | Lamarque et al. (2013) |
| | 79.56°W | 44.19°N | summer | 0.91 | Wu et al. (2016) |
| Amazon forest | 61.8°W | 10.1°S | wet | 1.1 | Rummel et al. (2007) |
| | 117.9°E | 4.9°N | wet | 1.0 | Fowler et al. (2011) |
| Coniferous forest | 3.4°W | 55.3°N | spring | 0.58 | Coe et al. (1995) |
| | 66.7°W | 54.8°N | summer | 0.26 | Munger et al. (1996) |
| | 11.1°E | 60.4°N | spring | 0.31 | Hole et al. (2004) |
| | | | summer | 0.48 | |
| | | | autumn | 0.2 | |
| | | | winter | 0.074 | |
| | 8.4°E | 56.3°N | spring | 0.68 | Mikkelsen et al. (2004) |
| | | | summer | 0.8 | |
| | | | autumn | 0.83 | |
| | 18.53°E | 49.55°N | Jul-Aug | 0.5 | Zapletal et al. (2011) |
| | 79.1°W | 36°N | spring | 0.79 | Finkelstein et al. (2000) |
| | 120.6°W | 38.9°N | summer | 0.59 | Kurpius et al. (2002) |
| | 0.7°W | 44.2°N | summer | 0.48 | Lamaud et al. (1994) |
| | 105.5°E | 40°N | summer | 0.39 | Turnipseed et al. (2009) |


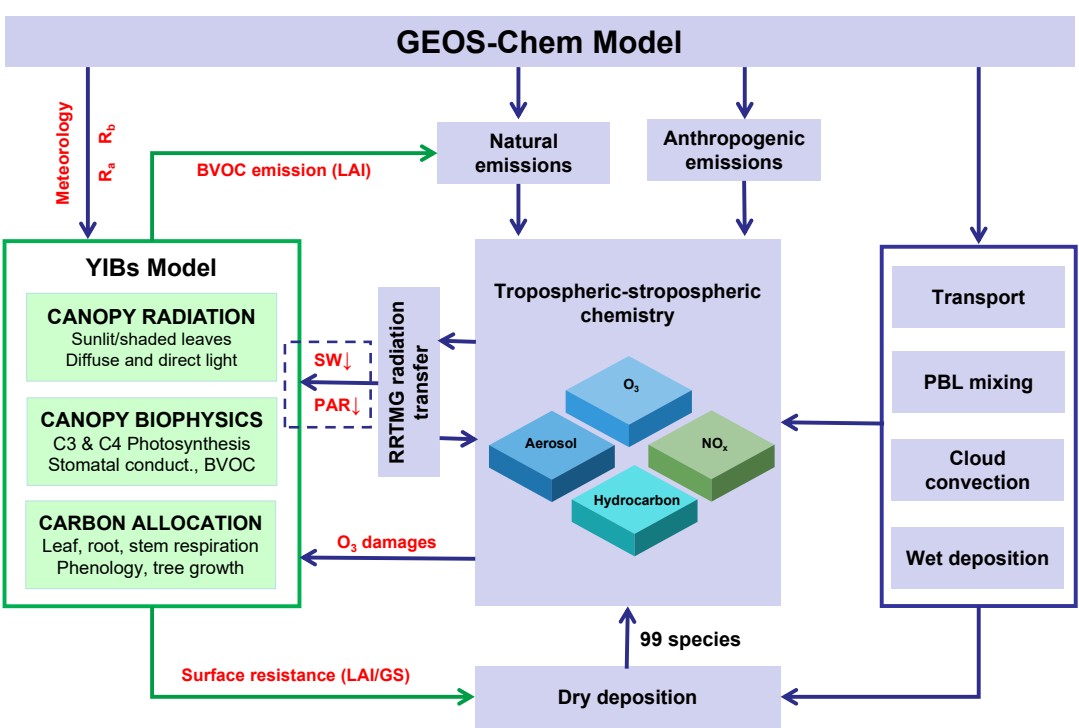

R$_a$: aerodynamic resistance; R$_b$: boundary layer resistance; GS: stomatal conductance; PAR: photosynthetically active radiation

**Figure 1** Diagram of the GC-YIBs global carbon-chemistry model. Processes with red fonts are implemented in this study. Processes with blue dashed box will be developed in the future.

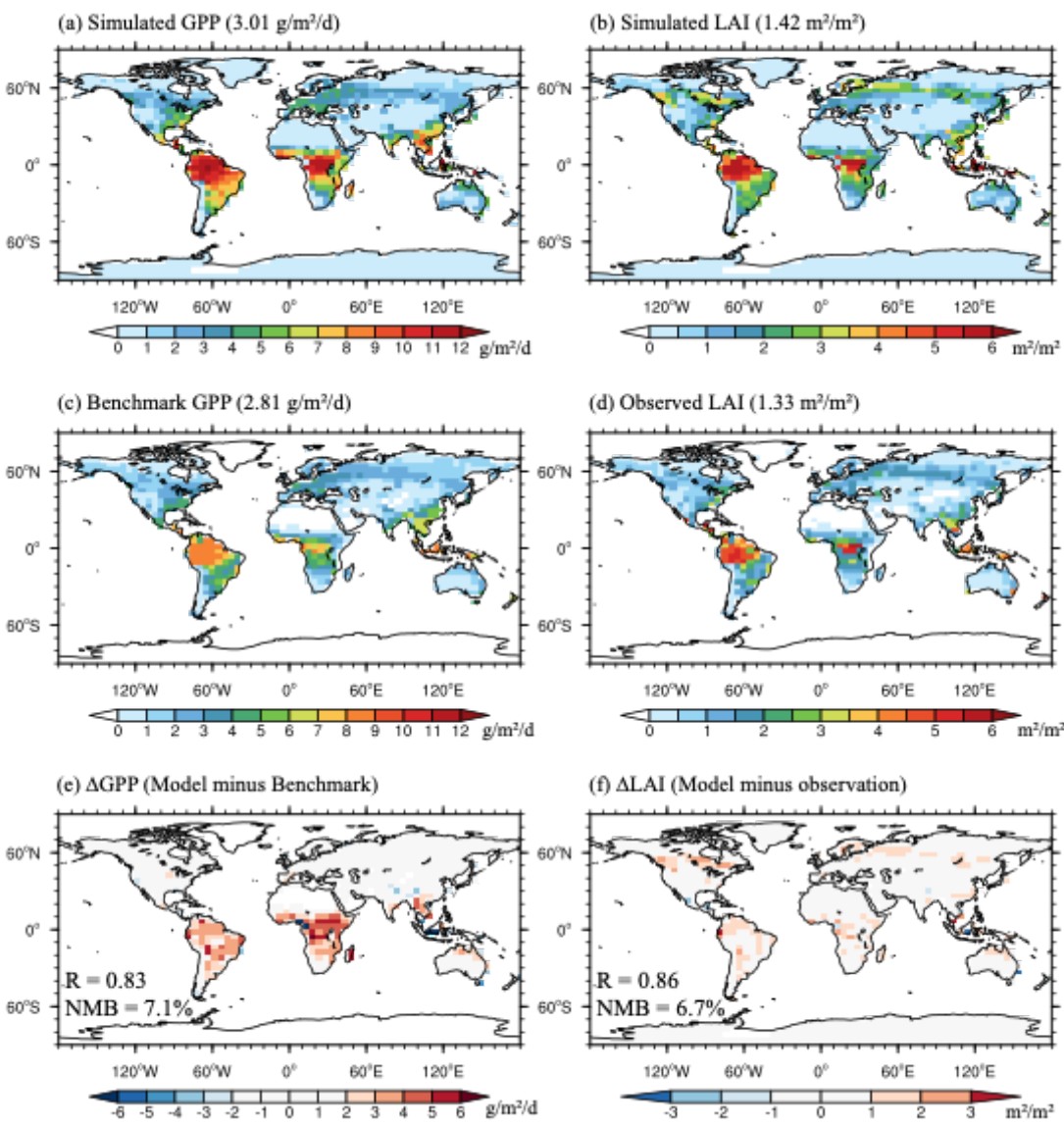

835

**Figure 2** Annual gross primary productivity (GPP) and leaf area index (LAI) from

Offline simulations **(a, b)**, observations **(c, d)**, and their differences **(e, f)** averaged for

period of 2010-2012. Global area-weighted GPP and LAI are shown on the title

brackets. The correlation coefficients (R) and global normalized mean biases (NMB)

are shown in the bottom figures.




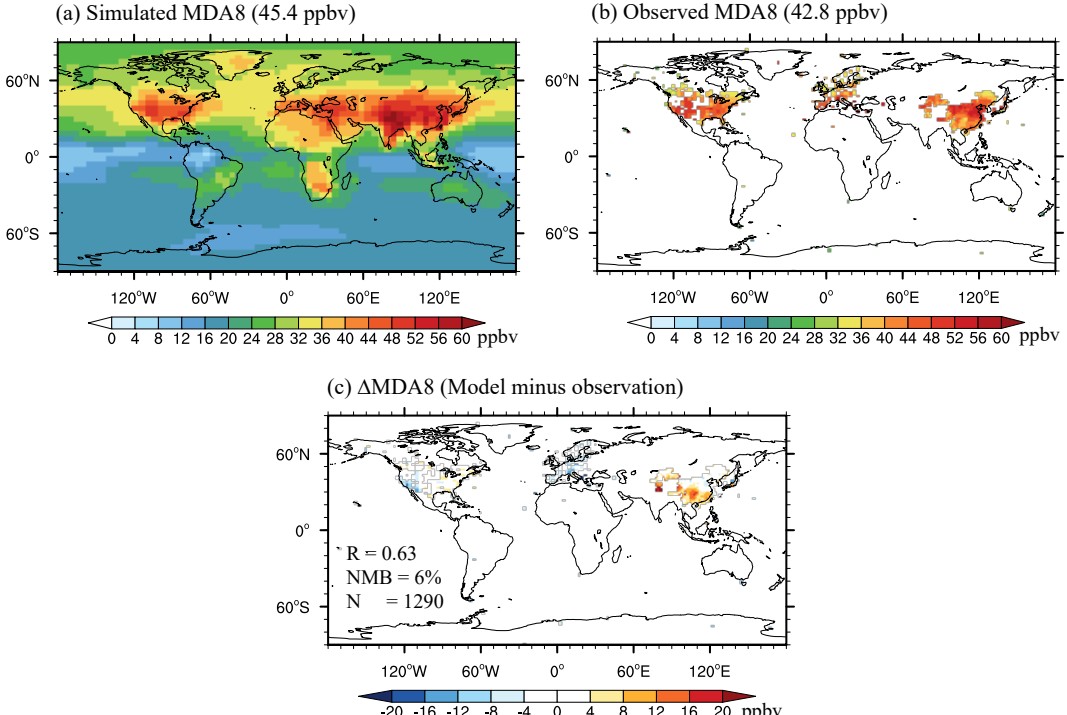


**Figure 3** Annual surface O$_3$ concentrations ([O$_3$]) from Offline simulations **(a)**,

observations **(b)**, and their differences **(c)** averaged for period of 2010-2012. Global

area-weighted surface [O$_3$] over grids with available observations are shown on the

title brackets. The correlation coefficient (R) and global normalized mean biases

(NMB) are shown in the bottom figures with indication of grid numbers (N) used for

statistics.




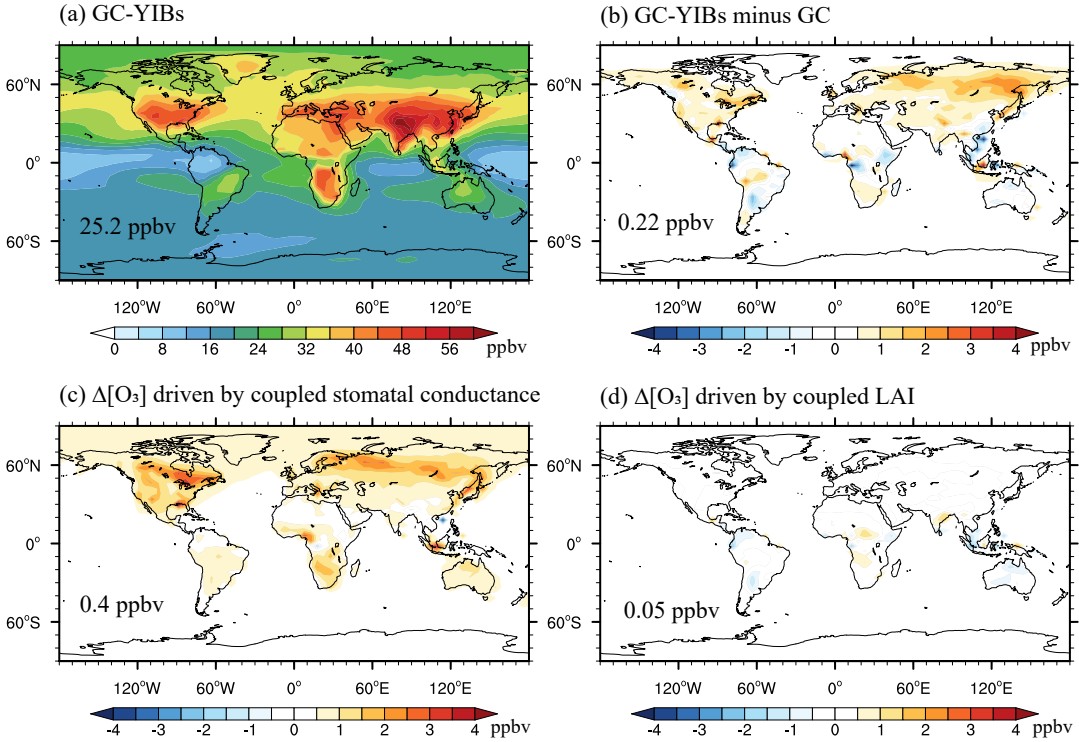


**Figure 4** Simulated annual surface [O₃] from online GC-YIBs model **(a)** and its

changes (b-d) relative to Offline simulations. Changes of [O₃] are caused by **(b)**

jointly coupled LAI and stomatal conductance (Online_ALL – Offline), **(c)** coupled

stomatal conductance alone (Online_ALL – Online_LAI), and **(d)** coupled LAI alone

(Online_ALL – Online_GS). Global area-weighted [O₃] or Δ[O₃] are shown in the

figures.



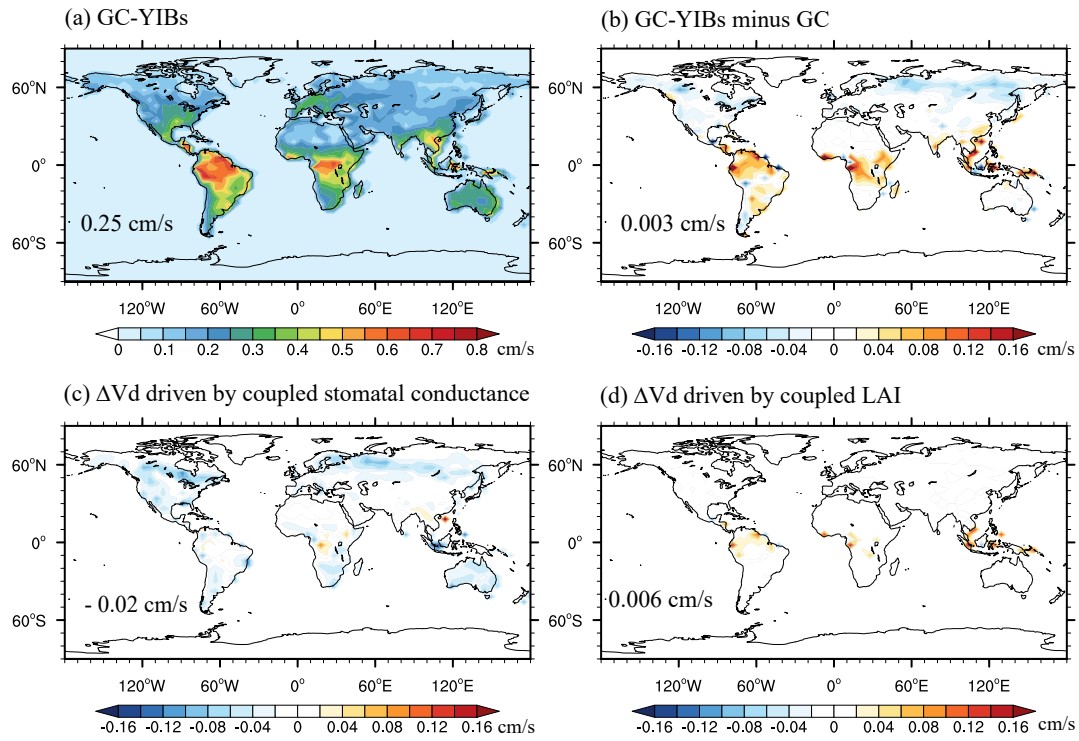


**Figure 5** Simulated annual O$_3$ dry deposition velocity from online GC-YIBs model **(a)** and its changes caused by coupled LAI and stomatal conductance **(b-d)** averaged for period of 2010-2012. The changes of dry deposition velocity are driven by **(b)** coupled LAI and stomatal conductance (Online_ALL – Offline), **(c)** coupled stomatal conductance alone (Online_ALL – Online_LAI), and **(d)** coupled LAI alone (Online_ALL – Online_GS). Global area-weighted annual O$_3$ dry deposition velocity and changes are shown in the figures.


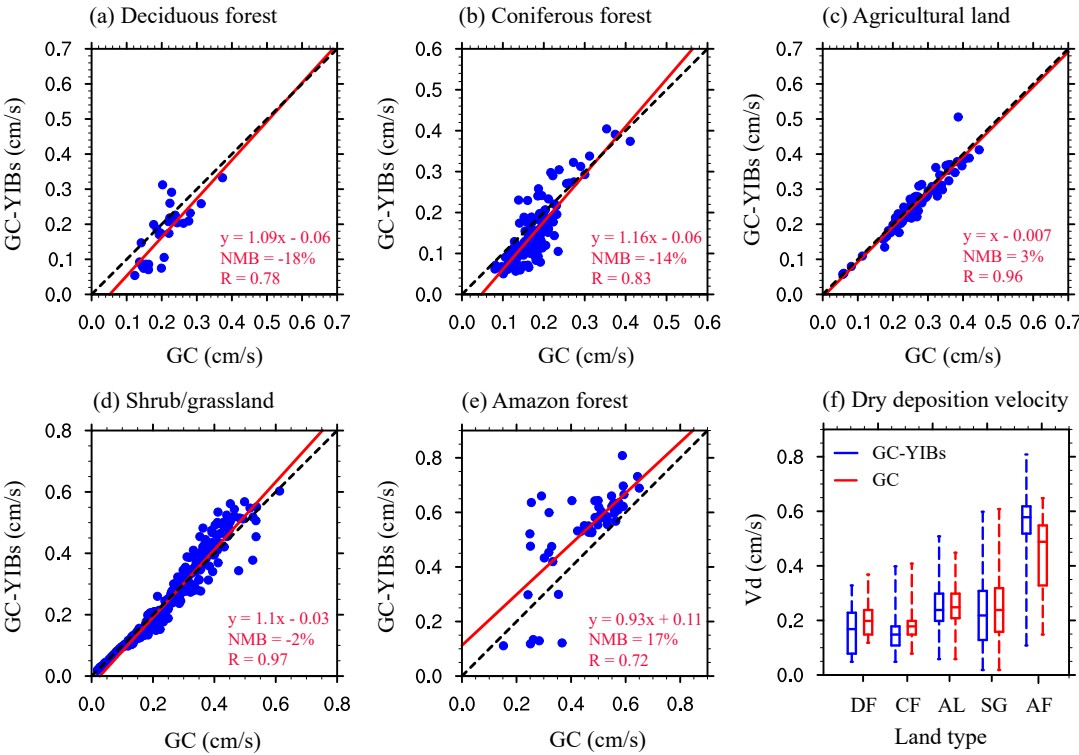


**Figure 6** Comparisons of annual O$_3$ dry deposition velocity between online GC-YIBs

(Online_ALL simulation) and GC (Offline simulation) models for different land types,

including **(a)** Deciduous forest, **(b)** Coniferous forest, **(c)** Agricultural land, **(d)**

Shrub/grassland, and **(e)** Amazon forest. The box plots of dry deposition velocity

simulated by online GC-YIBs (blue) and GC models (red) for different land types are

shown in **(f)**. Each point in (a)-(e) represents annual O$_3$ dry deposition velocity at one

grid point averaged for period of 2010-2012. The red lines indicate linear regressions

between predictions from GC-YIBs and GC models. The regression fit, correlation

coefficient (R), and normalized mean biases (NMB) are shown on each panel.

886

887

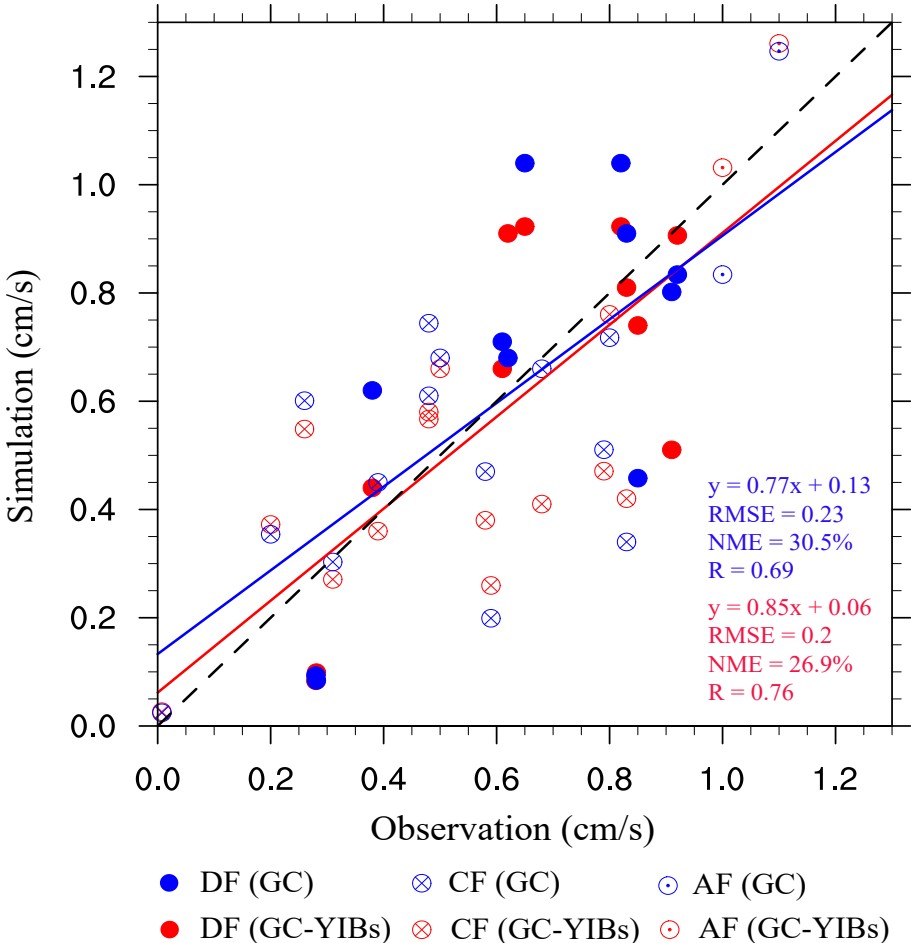

888

**Figure 7** Comparison between observed and simulated $O_3$ dry deposition velocity at

observational sites. The different marker types represent different land types. The blue

and red markers represent the simulation results from online GC-YIBs (Online_ALL

simulation) and GC (Offline simulation) models, respectively. The blue and red lines

indicate linear regressions between simulations and observations. The regression fits,

root-mean-square errors (RMSE), normalized mean errors (NME) and correlation

coefficients (R) for GC-YIBs (blue) and GC (red) models are also shown.

896

897

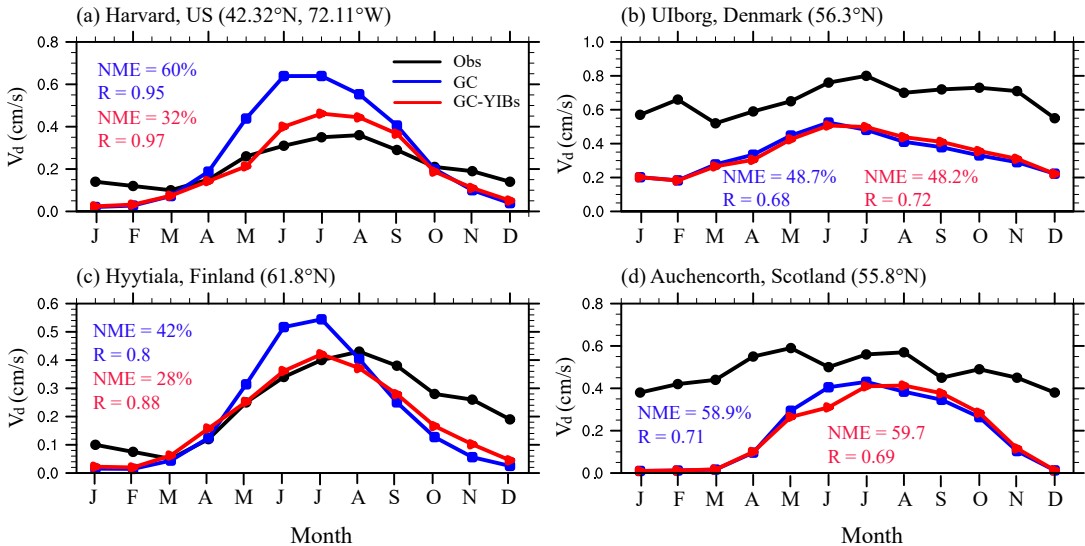

**Figure 8** Comparison of monthly O$_3$ dry deposition velocity at Harvard **(a)**, Ulborg **(b)**, Hyytiala **(c)** and Auchencorth **(d)** sites. The black lines represent observed O$_3$ dry deposition velocity. The blue and red lines represent simulated O$_3$ dry deposition velocity from GC (Offline simulation) and online GC-YIBs (Online_ALL simulation) models, respectively.

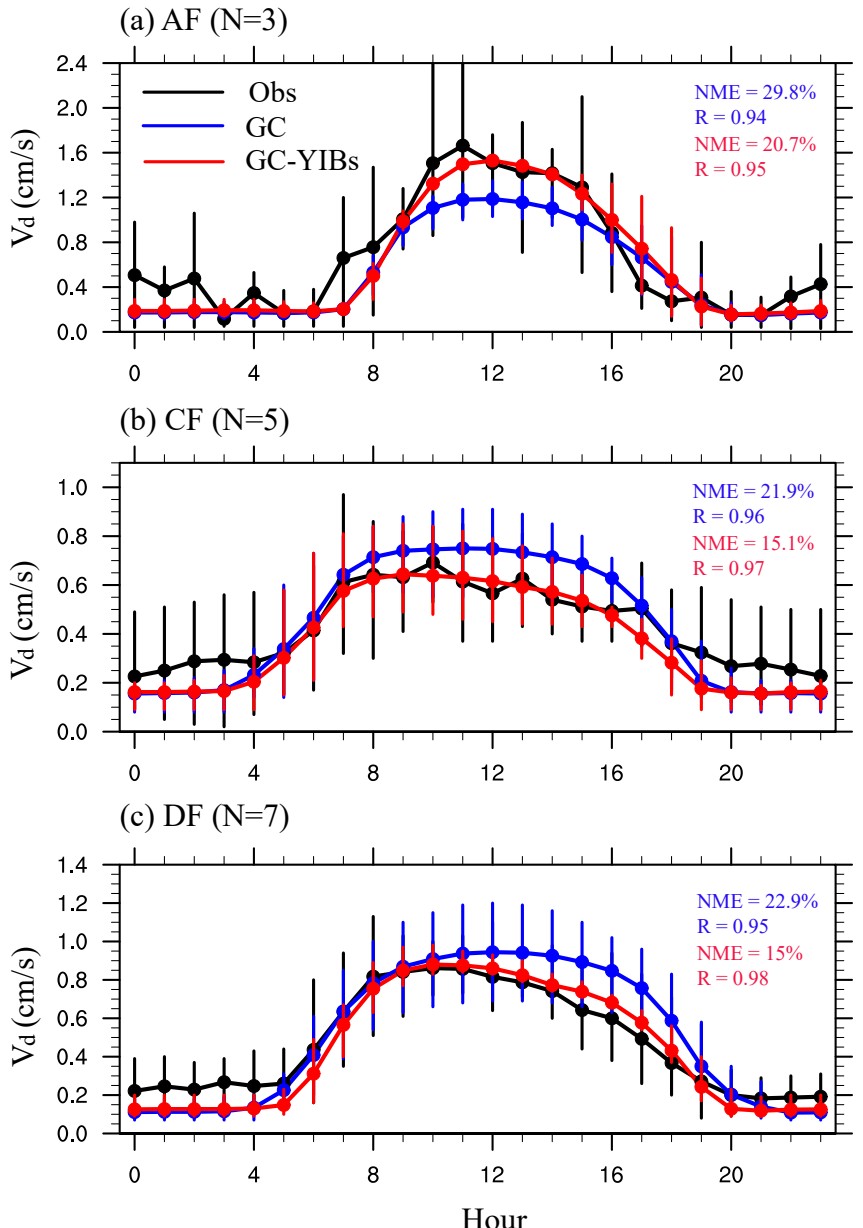

**Figure 9** Comparison of multi-site mean diurnal cycle of $O_3$ dry deposition velocity at Amazon **(a)**, coniferous **(b)** and deciduous **(c)** forests. Errorbars represent the range of values from different sites. Black lines represent observed $O_3$ dry deposition velocity. The blue and red lines represent simulated $O_3$ dry deposition velocity by GC (Offline simulation) and online GC-YIBs (Online_ALL simulation) models, respectively. The site number (N), R, and NME are shown for each panel.

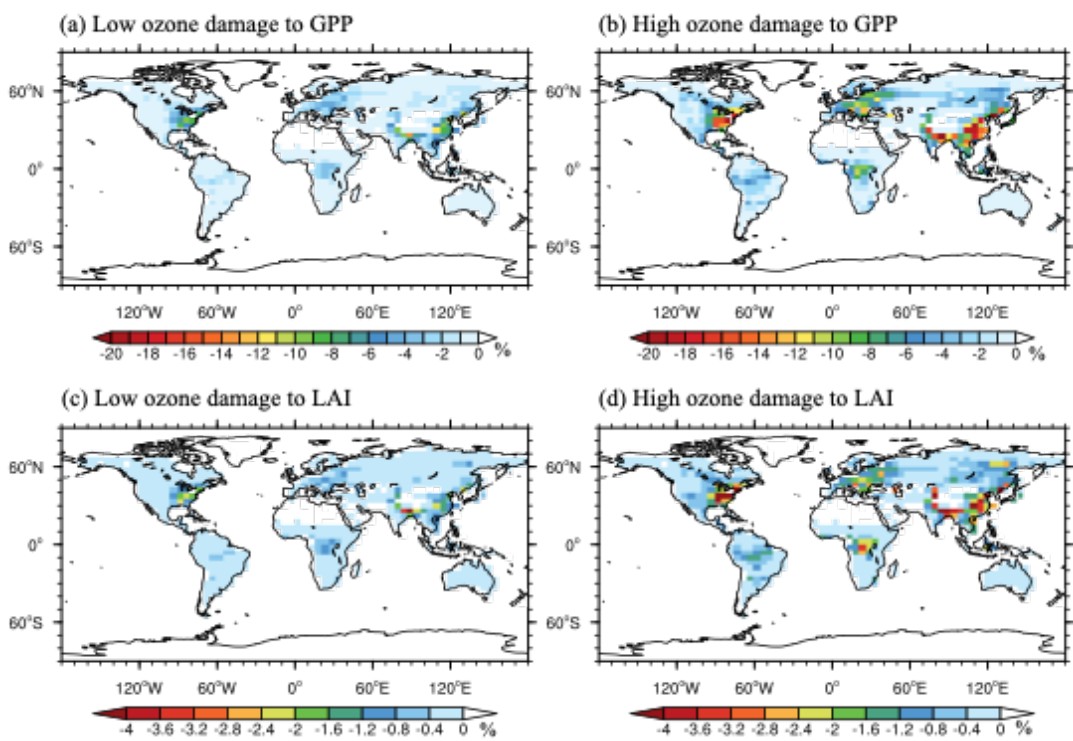

914

**Figure 10** Percentage changes in **(a, b)** GPP and **(c, d)** LAI caused by O₃ damaging

effects with **(a, c)** low (Online_ALL_LS simulation) and **(b, d)** high sensitivities

(Online_ALL_HS simulation). Both changes of GPP and LAI are averaged for 2010–

2012.

919

920