# Peer review of "Implementation of Yale Interactive terrestrial Biosphere model v1.0"

_Geoscientific Model Development, 2019_

## Referee Comment (RC1) · Anonymous Referee #1 · 25 Nov 2019

Review for Lei et al., "Implementation of Yale Interactive terrestrial Biosphere model version 1.0 into GEOS-Chem version 12.0.0: a tool for biosphere-chemistry interactions

REVIEW SUMMARY Lei et al., present a new model that combines a dynamic vegetation model that includes biogeochemistry (YIBs) with a widely used chemical transport model (GEOS-Chem). They run the model offline and with 5 different online conditions. They use model results to validate the model against measurements (particularly gross primary productivity and leaf area index). They explore the effects of building the online model on ozone mixing ratios, ozone deposition, and ozone damaging effects on terrestrial activity (such as gross primary productivity). In general, the global average change in ozone mixing ratios is quite small. However, they do find some notable differences in ozone deposition rates between GC and GC-YIBs, and they find the online model does improve ozone deposition rates when compared to the limited observations that are available. Finally, the utility of the model is demonstrated by their results on the effects of ozone on terrestrial productivity. Using the online model, they find gross primary productivity can decrease up to 15% in certain areas due to the damaging effects of ozone pollution. This study provides a valuable tool for investigating links between the terrestrial biosphere and atmospheric chemistry, which is a critical (and under-studied) research area for predicting the effects of climate change. The authors could improve the manuscript in a couple areas to better communicate their reasoning and clarify concepts to the reader. I recommend the paper for publication after addressing the minor comments summarized below, which should help them accomplish this.

SPECIFIC COMMENTS Section 3.2, particularly lines 305-308. The authors state that the difference in ozone mixing ratios between the Online_All and Online_LAI suggests that "changes in stomatal conductance play the dominant role in regulating surface [O3]." I am not following this logic and I think they need to better clarify how they are making this connection. The description of the model runs just says Online_All has daily dynamically predicted LAI and hourly predicted stomatal conductance while the Online_LAI has daily dynamically predicted LAI and the original dry deposition scheme. It is not obvious to me how comparing the output of these two model simulations leads to the conclusion they have provided, and this could be better explained.

Discussion of Figure 6 and 7: it is unclear what value is added by including figure 6. The figure shows the different land types in the original GC dry deposition scheme where different land types are prescribed a fixed parameters for stomatal conductance. The online model is different because it calculates stomatal conductance based on photo-

[Figure]

synthesis and environmental forcings (L. 332-333). Then they show that dry deposition comparisons between the original and online model vary by biome type in Figure 7. This would be expected simply knowing the original model uses prescribed parameters based on land type while the online model calculates stomatal conductance! The map shown in Figure 6 does not provide any additional useful information. It might be more helpful to describe in more detail how the fixed parameters in the original GC model were developed. That would be more useful than the map of different land types.

Figure 7: it is unclear which online GC-YIBs conditions were used to generate this figure. Five different online conditions were described in the methods and it should be clarified for each figure which model results are being included. In general, the authors do a good job making this clear, but Fig 7 stands out as an example where they did not specify this.

TECHNICAL CORRECTIONS

Page 13, L. 284: missing a period at the end of the last sentence

Page 14, L. 290: "[. . .] model overestimates annual [O3] in southern China while predicts lower values in western Europe [. . .]". "while predicts" is not the correct grammar.

Page 14, L. 300: "GC-YIBs predicts larger [O3] of 0.5-2 ppbv". I think the authors mean the GC-YIBs predicts HIGHER [O3] BY 0.5-2 ppbv.

---

## Referee Comment (RC2) · Anonymous Referee #2 · 9 Dec 2019

This study represents a new biosphere-chemistry modeling framework that simulates online, two-way interactions between surface ozone and vegetation, mainly through the linkages between stomatal conductance, leaf area index (LAI) and dry deposition. Global model-observation comparison for simulated gross primary productivity (GPP), LAI, ozone concentrations and dry deposition velocities have been conducted using a large ensemble of datasets. This work is important in laying a foundation for more in-depth future studies of biosphere-atmosphere interactions. However, as of the current form the manuscript lacks sufficient details regarding model implementation, which

[Figure]

I believe is important for a GMD paper. I would recommend the publication of this manuscript should the following model details are included, addressed and discussed.

Specific comments:

P6 L129: I think here and elsewhere, the units for all variables should be included in all the equations listed.

P7 L137: Carbon allocation and LAI simulations are a very important part of the modeling framework, but no details have been given. The schemes/algorithms used for simulating carbon allocation and LAI should be described.

P8 L157: Why is aerodynamic resistance not included in the calculation of ozone fluxes? The ozone simulated by any chemical transport model should be at the lowest model layer, but that should be different enough from the ozone concentration at the canopy top. Please justify. Moreover, shouldn't the ozone flux calculated here for ozone damage be consistent with the dry deposition velocity/flux calculation in GC? The internally inconsistent ways to represent ozone fluxes between GC and YIBs seem to reduce the usefulness of GC-YIBs as a coupling tool.

P8 L168: $4° \times 5°$ appears to be a rather low resolution. While the issue of computational expense is understandable, I recommend the authors to discuss how such a low resolution of simulations may interfere with the accuracy of simulated variables (ozone concentrations, GPP, etc.) as compared with observations.

P10 L210: While the replacement of Olson land-type stomatal resistance with YIBs plant-functional-type (PFT) stomatal resistance is mentioned, could the authors also explain how the conversion of other land-type resistances to YIBs PFT resistances was done? In general, it would be highly useful to explain how Olson land types are matched and mapped with YIBs PFTs. A conversion table in the supplement would really help.

P11 L223: YIBs simulates stomatal conductance first at the leaf level, while GC takes

in conductance at the canopy level. Appropriate scaling between the two levels should be included and discussed.

P12 L250: Four years of spin-up for LAI simulations is probably insufficient. LAI typically takes decades to stabilize, depending on the initial conditions of LAI. The authors are recommended to explain in greater detail such an issue, show whether LAI has reached a steady state in four years, and state specifically what LAI is used as the initial conditions.

P14 L306: I think the authors meant Online_GS here instead of Online_LAI.

P15 L308: I think the authors meant Online_LAI here instead of Online_GS.

P15 L324: The authors need to justify why BVOC changes resulting from LAI changes are not the dominant factor (in addition to stating the broadly consistent spatial patterns). How BVOC changes should influence the results and interpretation should be discussed in greater detail.

---

## Referee Comment (RC3) · Anonymous Referee #3 · 25 Dec 2019

This work integrates and couples together a global atmospheric chemistry model (GEOS-Chem) and a terrestrial biosphere model (YIBs) in order to investigate the feedbacks associated between the two, often separately simulated, systems. First, the authors evaluate their integrated model against observed or baseline measures of plant activity (GPP/LAI) and an example chemical species (ozone concentration). They also compare the performance of the coupled and integrated models against observed ozone dry deposition velocities, finding the coupled model an improvement. Using this coupled model, the authors then investigate the impact ozone concentration has on

plant activity using differing sensitivities to ozone damage. Overall, this work is timely and addresses an important issue within the modeling of these systems. The description of the model and evaluation is carried-out well with appropriately supportive figures. However, the paper does not go far enough to be truly impactful and confidently useful to the community in its present form, but rather, substantial addition and expansion is required for publishing in GMD. The authors should either expand the evaluation of the model to show that coupling truly does improve comparisons or provide additional applicational evidence for the importance of such coupling to understanding biosphere-atmosphere interactions. Further specific comments and recommendations are listed below.

1) While the PM impact on plants is mentioned as an important process to consider in the introduction (lines 63-69), there is no integration description or evaluation in this paper, and no further mention until the last paragraph. Perhaps clarify the focus of the paper at the beginning to adjust expectations.

2) Aerosols are not always beneficial to vegetation if the total radiation decreases more than the enhancing effect caused by diffusion (line 64).

3) Since GC-YIBs integrates two existing models, sections 2.1 and 2.2 can be trimmed to only include the relevant equations and processes discussed in the remainder of the paper.

4) More description of the "satellite-based land types and cover fraction" (lines 122 and 229) would be useful as this is quite vague.

5) The fact that coefficient a is uncertain and can and will be varied in different simulations is not clear from the current description in line 153.

6) Much work has been done to evaluate the GEOS-Chem dry deposition scheme for ozone and understand the importance of dry deposition schemes in general (e.g. Silva and Heald 2018, JGR, Wong et al 2019, ACP) but these issues are not mentioned

here (neither sections 2.2 nor 4). Especially important to consider is lack of observations to truly constrain ozone dry deposition globally and the uncertainty over various timescales and in spatially heterogenous regions.

7) The title of section 2.5 should read "Evaluation data", as models are evaluated, not validated.

8) Why are only 9 sites used for the comparison of ozone dry deposition velocity (lines 266, 341-355, Table 2, Figure 8)? Many more data are available as in Silva and Heald 2018.

9) Further description of the limitations and errors of both the observed LAI and GPP product should be included (section 2.5), and clarification should be made that GPP is not observed (line 271).

10) How do the simulated GPP/LAI and ozone concentrations from offline GC-YIBs compare to those values from the original YIBs and GC, respectively? Are the original model configurations degraded or enhanced by the integration and use of a common land type and meteorological driver? Are the magnitudes of these changes similar to the noted improvements seen when the coupling is turned on?

11) Line 281 attributes the GPP bias to an underestimation in the benchmark GPP for tropical rainforest. Could the differences from using a different meteorology dataset instead be biasing the model (line 283)?

12) Compared to what other drivers (BVOC emissions changes?) are dry deposition velocities the dominant driver in the change in O3 (line 324)? Try testing the impacts of the changing other drivers, rather than relying only on consistent spatial patterns (line 323).

13) Given the small sample size and scattered data (Figure 8), the statistics cited for the comparison of dry deposition velocities in coupled GC-YIBs compared to offline GC-YIBs do not provide for high confidence that the model is truly improved with the

coupling of these systems (lines 341-355). A more robust analysis should be undertaken to account for the errors in both the observed and simulated values and present the confidence with which the model could be said to truly be improved.

14) The coupling of these systems for the assessment of ozone damages to vegetation is presented as a key motivation for this study, but the differences in damage between this coupled model and previous models are not discussed (mentioned only in line 372). The discussion should be expanded to explain the differences and highlight the advantages of coupling the systems in section 3.3.

15) Other studies including Lin et al, 2019 GBC for the GFDL models have also investigated the coupled biosphere and atmosphere in similar ways with regards to ozone and are worth discussion in addition to the CESM work. If the ozone dry deposition is the chief application of the model so far, more clarity should be made in the discussion of the uncertainties that already exist in simulating dry deposition globally.

16) One way to justify the slow model speed (line 420) for the modest model improvements shown through coupling would be to expand upon the usefulness of the applications only so far mentioned in lines 428-444.

17) While supported in part at Harvard, GEOS-Chem is developed and maintained by a global community of atmospheric chemists, not one group (line 449), and should be acknowledged as such.

18) Minor grammatical issues are present throughout, especially omission of articles before nouns. (example, line 48 "from terrestrial biosphere").

Papers cited: Lin et al., Sensitivity of Ozone Dry Deposition to Ecosystem–Atmosphere Interactions: A Critical Appraisal of Observations and Simulations, https://doi.org/10.1029/2018GB006157, (2019). Silva and Heald, Investigating Dry Deposition of Ozone to Vegetation, https://doi.org/10.1002/2017JD027278, (2018). Wong et al., Importance of dry deposition parameterization choice in global simulations of

surface ozone, https://doi.org/10.5194/acp-19-14365-2019, (2019).
* * *

---

## Author Comment (AC1) · 20 Jan 2020

**Response to the reviewer 1**

We are grateful to the referees for their time and energy in providing helpful comments and guidance that have improved the manuscript. In this document, we describe how we have addressed the reviewer's comments. Referee comments are shown in black and author responses are shown in blue text.

**Review summary:**

Lei et al., present a new model that combines a dynamic vegetation model that includes biogeochemistry (YIBs) with a widely used chemical transport model (GEOS-Chem). They run the model offline and with 5 different online conditions. They use model results to validate the model against measurements (particularly gross primary productivity and leaf area index). They explore the effects of building the online model on ozone mixing ratios, ozone deposition, and ozone damaging effects on terrestrial activity (such as gross primary productivity). In general, the global average change in ozone mixing ratios is quite small. However, they do find some notable differences in ozone deposition rates between GC and GC-YIBs, and they find the online model does improve ozone deposition rates when compared to the limited observations that are available. Finally, the utility of the model is demonstrated by their results on the effects of ozone on terrestrial productivity. Using the online model, they find gross primary productivity can decrease up to 15% in certain areas due to the damaging effects of ozone pollution. This study provides a valuable tool for investigating links between the terrestrial biosphere and atmospheric chemistry, which is a critical (and under-studied) research area for predicting the effects of climate change. The authors could improve the manuscript in a couple areas to better communicate their reasoning and clarify concepts to the reader. I recommend the paper for publication after addressing the minor comments summarized below, which should help them accomplish this.

➔ Thank you for your positive evaluations. All the questions and concerns have been

carefully answered and the paper has been revised accordingly.

**Specific comments:**

Section 3.2, particularly lines 305-308. The authors state that the difference in ozone mixing ratios between the Online_All and Online_LAI suggests that "changes in stomatal conductance play the dominant role in regulating surface $[O_3]$." I am not following this logic and I think they need to better clarify how they are making this connection. The description of the model runs just says Online_All has daily dynamically predicted LAI and hourly predicted stomatal conductance while the Online_LAI has daily dynamically predicted LAI and the original dry deposition scheme. It is not obvious to me how comparing the output of these two model simulations leads to the conclusion they have provided, and this could be better explained.

*Response*: The configurations of Online_ALL and Online_LAI simulations are the same except for stomatal conductance. Online_ALL simulation uses hourly stomatal conductance simulated by YIBs, which dynamically responds to environmental factors (e.g., temperature, water stress, radiation, $CO_2$ and so on). However, Online_LAI simulation uses prescribed stomatal conductance, though it uses online-predicted LAI the same as Online_ALL. As a result, the difference between Online_ALL and Online_LAI represents the effects of updated stomatal conductance on surface $[O_3]$. In revised paper, we changed "the original dry deposition scheme" to "prescribed stomatal conductance" to clarify. (Line 298)

Discussion of Figure 6 and 7: it is unclear what value is added by including figure 6. The figure shows the different land types in the original GC dry deposition scheme where different land types are prescribed fixed parameters for stomatal conductance. The online model is different because it calculates stomatal conductance based on photo-synthesis and environmental forcings (L. 332-333). Then they show that dry deposition comparisons between the original and online model vary by biome type in Figure 7. This would be expected simply knowing the original model uses prescribed

parameters based on land type while the online model calculates stomatal conductance! The map shown in Figure 6 does not provide any additional useful information. It might be more helpful to describe in more detail how the fixed parameters in the original GC model were developed. That would be more useful than the map of different land types.

*Response*: The main purpose of Fig. 6 is to show the location and deposition land type of sites (black points) used for evaluations of dry deposition. We have moved Fig.6 into SI as suggested (now Fig. S2).

[Figure]

**Figure S2** The major dry deposition land type at each grid cell converted from YIBs land types. DF, CF, AL, SG and AF represent deciduous forest, coniferous forest, agricultural land, shrub/grassland and amazon forest, respectively. Black dots indicate the locations of measurement sites used in evaluation (Table 2).

Figure 7: it is unclear which online GC-YIBs conditions were used to generate this figure. Five different online conditions were described in the methods and it should be clarified for each figure which model results are being included. In general, the authors do a good job making this clear, but Fig 7 stands out as an example where they did not specify this.

*Response*: Results of online GC-YIBs shown in Figure 7 (now Figure 6) are from simulation Online_ALL. In the revised paper, we clarified in the figure caption as follows: "Figure 6 Comparisons of annual $O_3$ dry deposition velocity between online GC-YIBs (Online_ALL simulation) and GC (Offline simulation) models for different land types …"

**Technical corrections:**

Page 13, L. 284: missing a period at the end of the last sentence

*Response*: Corrected as suggested.

Page 14, L. 290: "[: : :] model overestimates annual $[O_3]$ in southern China while predicts lower values in western Europe [: : :]". "while predicts" is not the correct grammar.

*Response*: We revised the sentence as follows: "Although offline GC-YIBs model overestimates annual [O3] in southern China and predicts lower values in western Europe and western U.S." (Lines 346-347)

Page 14, L. 300: "GC-YIBs predicts larger $[O_3]$ of 0.5-2 ppbv". I think the authors mean the GC-YIBs predicts HIGHER $[O_3]$ BY 0.5-2 ppbv.

*Response*: Corrected as suggested.

---

## Author Comment (AC2) · 20 Jan 2020

**Response to the reviewer 2**

We are grateful to the referees for their time and energy in providing helpful comments and guidance that have improved the manuscript. In this document, we describe how we have addressed the reviewer's comments. Referee comments are shown in black and author responses are shown in blue text.

This study represents a new biosphere-chemistry modeling framework that simulates online, two-way interactions between surface ozone and vegetation, mainly through the linkages between stomatal conductance, leaf area index (LAI) and dry deposition. Global model-observation comparison for simulated gross primary productivity (GPP), LAI, ozone concentrations and dry deposition velocities have been conducted using a large ensemble of datasets. This work is important in laying a foundation for more indepth future studies of biosphere-atmosphere interactions. However, as of the current form the manuscript lacks enough details regarding model implementation, which I believe is important for a GMD paper. I would recommend the publication of this manuscript should the following model details are included, addressed and discussed.

Thank you for your positive evaluations. All the questions and concerns have been carefully answered and the paper has been revised accordingly.

**Specific comments:**

P6 L129: I think here and elsewhere, the units for all variables should be included in all the equations listed.

*Response*: Units for all equation variables have been added in the revised paper. For Equation (1), we described it as follows: "... where  $r_s$  is the leaf stomatal resistance  $(s m^{-1})$ ; *m* is the empirical slope of the Ball-Berry stomatal conductance equation and is affected by water stress;  $c_s$  is the CO2 concentration at the leaf surface  $(\mu mol m^{-3})$ ; *RH* is the relative humidity of atmosphere; *b*  $(m s^{-1})$  represents the

minimum leaf stomatal conductance when net leaf photosynthesis  $(A_{net}, \mu mol \ m^{-2} \ s^{-1})$  is 0." (Lines 147-151)

P7 L137: Carbon allocation and LAI simulations are a very important part of the modeling framework, but no details have been given. The schemes/algorithms used for simulating carbon allocation and LAI should be described.

*Response*: In the revised paper, we added following descriptions in section 2.1 "Descriptions of the YIBs model" to clarify (Lines 165-184):

The YIBs model applies the LAI and carbon allocation schemes from the TRIFFID model (Clark et al., 2011; Cox, 2001). On the daily scale, canopy LAI is calculated as follows:

$$LAI = f \times LAI_{max} \tag{3}$$

Where f represents phenological factor controlled by meteorological variables (e.g., temperature, water availability, and photoperiod);  $LAI_{max}$  represents the available maximum LAI related to tree height, which is dependent on the vegetation carbon content ( $C_{veg}$ ). The  $C_{veg}$  is calculated as follows:

$$C_{veg} = C_l + C_r + C_w \tag{4}$$

where  $C_l$ ,  $C_r$  and  $C_w$  represent leaf, root, and stem carbon contents, respectively. And all carbon components are parameterized as the function of  $LAI_{max}$ :

$$\begin{cases} C_l = \alpha \times LAI \\ C_r = \alpha \times LAI_{max} \\ C_r = \beta \times LAI_{max}^{\gamma} \end{cases}$$
(5)

where  $\alpha$  represents the specific leaf carbon density;  $\beta$  and  $\gamma$  represent allometric parameters. The vegetation carbon content  $C_{veg}$  is updated every 10 days:

$$\frac{dC_{veg}}{dt} = (1 - \tau) \times NPP - \varphi \tag{6}$$

where  $\tau$  and  $\varphi$  represent partitioning parameter and litter fall rate, respectively, and their calculation methods have been documented in Yue and Unger (2015). Net primary productivity (*NPP*) is calculated as the residue of subtracting autotrophic respiration ( $R_a$ ) from GPP:

$$NPP = GPP - R_a \tag{7}$$

P8 L157: Why is aerodynamic resistance not included in the calculation of ozone fluxes? The ozone simulated by any chemical transport model should be at the lowest model layer, but that should be different enough from the ozone concentration at the

canopy top. Please justify. Moreover, shouldn't the ozone flux calculated here for ozone damage be consistent with the dry deposition velocity/flux calculation in GC? The internally inconsistent ways to represent ozone fluxes between GC and YIBs seem to reduce the usefulness of GC-YIBs as a coupling tool.

*Response*: Thank you for the constructive comments.

(i) The GEOS-Chem model calculates concentrations of air components at 47 vertical layers from 1013.25 to 0.01 hPa. We use  $[O_3]$  at the lowest layer to approximate  $O_3$  concentration at the canopy top. We acknowledge the limit of this approximation in the discussion section: "(3)  $[O_3]$  at the lowest model level is used as an approximation of canopy  $[O_3]$ . The current model does not include a sub-grid parameterization of pollution transport within canopy, leading to biases in estimating  $O_3$  vegetation damage and the consequent feedback. However, development of such parameterization is limited by the availability of simultaneous measurements of microclimate and air pollutants." (Lines 506-510)

(ii) In the original YIBs model,  $O_3$  stomatal flux is calculated as the function of boundary layer resistance, stomatal resistance, and ambient  $O_3$  concentration. In order to fully link GEOS-Chem with YIBs, we have updated the stomatal  $O_3$  flux scheme to include aerodynamic resistance (which is now consistent with GEOS-Chem). At each integration step, GEOS-Chem provides both hourly aerodynamic resistance ( $r_a$ ) and boundary resistance ( $r_b$ ) for stomatal  $O_3$  flux scheme in YIBs.

In the revised paper, Eq.9  $F_{O_3} = \frac{[O_3]}{r_b + k \cdot r_s}$  has been updated as  $F_{O_3} = \frac{[O_3]}{r_a + r_b + k \cdot r_s}$  to take into effects of both  $r_a$  and  $r_b$ . We clarify in the revised paper as follows: "In the online GC-YIBs configuration, GC provides the hourly meteorology, aerodynamic resistance, boundary layer resistance, and surface  $[O_3]$  to YIBs." (Lines 259-261). Accordingly, the assessment of global O3 damage to vegetation (section 3.3) has been updated. Compared to the original stomatal O3 flux scheme within YIBs, the new scheme increases O3 stomatal flux in Amazon but decreases O3 stomatal flux in eastern China (Fig.R1c). As a result, O3 damage on GPP decreases in eastern China but increases in Amazon (Fig.R1f).

**Figure R1** Comparison of  $O_3$  stomatal flux schemes. (a) and (b) represent the  $O_3$  stomatal flux for new and original schemes, respectively. (c) represents the  $O_3$  stomatal flux difference between new and original schemes (a-b). (d) and (e) represent the  $O_3$  damages to GPP for new and original schemes, respectively. (f) represents the  $O_3$  damages difference between new and original schemes (d-e).

P8 L168:  $4^{\circ} \times 5^{\circ}$  appears to be a rather low resolution. While the issue of computational expense is understandable, I recommend the authors to discuss how such a low resolution of simulations may interfere with the accuracy of simulated variables (ozone concentrations, GPP, etc.) as compared with observations.

*Response*: We run relatively high resolution  $(2^{\circ} \times 2.5^{\circ})$  of GC-YIBs from 2006 to 2007 to have a check. The result of 2007 is used to compare the differences induced by resolutions. The following information has been added in the last part:

"The low resolution will affect local emissions (e.g., NOx and VOC) and transport, leading to changes in surface  $[O_3]$  in GEOS-Chem. The comparison results of 2007 show that low resolution of 4°×5° induces a global mean bias of -0.24 ppbv on surface  $[O_3]$  compared to the relatively high resolution at 2°×2.5° (Fig. S7). Compared with surface  $[O_3]$ , low resolution causes limited differences in vegetation variables (e.g., GPP and LAI, not shown)." (Lines 515-521).

---

## Author Comment (AC3) · 20 Jan 2020

**Response to the reviewer 3**

We are grateful to the referees for their time and energy in providing helpful comments and guidance that have improved the manuscript. In this document, we describe how we have addressed the reviewer's comments. Referee comments are shown in black and author responses are shown in blue text.

This work integrates and couples together a global atmospheric chemistry model (GEOS-Chem) and a terrestrial biosphere model (YIBs) in order to investigate the feedbacks associated between the two, often separately simulated, systems. First, the authors evaluate their integrated model against observed or baseline measures of plant activity (GPP/LAI) and an example chemical species (ozone concentration). They also compare the performance of the coupled and integrated models against observed ozone dry deposition velocities, finding the coupled model an improvement. Using this coupled model, the authors then investigate the impact ozone concentration has on plant activity using differing sensitivities to ozone damage. Overall, this work is timely and addresses an important issue within the modeling of these systems. The description of the model and evaluation is carried-out well with appropriately supportive figures. However, the paper does not go far enough to be truly impactful and confidently useful to the community in its present form, but rather, substantial addition and expansion is required for publishing in GMD. The authors should either expand the evaluation of the model to show that coupling truly does improve comparisons or provide additional applicational evidence for the importance of such coupling to understanding biosphere-atmosphere interactions. Further specific comments and recommendations are listed below.

**Thank you for your positive evaluations. All the questions and concerns have been carefully answered and the paper has been revised accordingly.**

1) While the PM impact on plants is mentioned as an important process to consider in

the introduction (lines 63-69), there is no integration description or evaluation in this paper, and no further mention until the last paragraph. Perhaps clarify the focus of the paper at the beginning to adjust expectations.

*Response*: We aim to develop a fully coupled biosphere-chemistry model GC-YIBs. We clarify in the introduction section that: "For the first step, we focus on the coupling between  $O_3$  and vegetation. The interactions between aerosols and vegetation will be developed and evaluated in the future." (Lines 113-115) The aerosols-vegetation interaction has been marked with blue dashed box in Fig. 1.

2) Aerosols are not always beneficial to vegetation if the total radiation decreases more than the enhancing effect caused by diffusion (line 64).

*Response*: The effect of aerosols on vegetation has been modified as following: "Unlike  $O_3$ , the effect of aerosols on vegetation is dependent on the aerosol concentrations. Moderate increase of aerosols in the atmosphere is beneficial to vegetation (Mahowald, 2011; Schiferl and Heald, 2018). The aerosol-induced enhancement in diffuse light results in more radiation reaching surface from all directions than solely from above. As a result, leaves in the shade or at the bottom can receive more radiation and are able to assimilate more  $CO_2$  through photosynthesis, leading to an increase of canopy productivity (Mercado et al., 2009; Yue and Unger, 2018). However, excessive aerosol loadings reduce canopy productivity because the total radiation is largely weakened (Alton, 2008; Yue and Unger, 2017)." (Lines 64-73)

3) Since GC-YIBs integrates two existing models, sections 2.1 and 2.2 can be trimmed to only include the relevant equations and processes discussed in the remainder of the paper.

*Response*: Thank you for your suggestion. Reviewer#2 expects us to add more details about the YIBs model. We have described some important processes within YIBs (e.g., the method calculating LAI, equations 3-7). These descriptions are especially useful to those unfamiliar with the YIBs model.

4) More description of the "satellite-based land types and cover fraction" (lines 122 and 229) would be useful as this is quite vague.

*Response*: We added Fig. S1 to show the land types used in YIBs. In addition, the conversion relationships between YIBs and GEOS-Chem deposition land types has been added (Table S2 and Fig. S2).

Figure S1 Fractional coverage of each land type at each grid cell.

5) The fact that coefficient  $\alpha$  is uncertain and will be varied in different simulations is not clear from the current description in line 153.

*Response*: We added Table S1 to clarify: "For a specific PFT, the values of coefficient *a* vary from low to high to represent a range of uncertainties for ozone vegetation damaging (Table S1)." (Lines 192-193)

| PFTs                        | $\alpha$ for high sensitivity     | $\alpha$ for low sensitivity       |
|-----------------------------|-----------------------------------|------------------------------------|
|                             | $(\text{mmol}^{-1}\text{m}^{-2})$ | $(\text{mmol}^{-1}\text{ m}^{-2})$ |
| Evergreen broadleaf forest  | 0.15                              | 0.04                               |
| Evergreen needleleaf forest | 0.075                             | 0.02                               |
| Deciduous broadleaf forest  | 0.15                              | 0.04                               |
| Shrub                       | 0.1                               | 0.03                               |
| Tundra                      | 0.1

---

## Author Response (AR2)

*Comments to the Author:*

*The paper is scientifically ready for final publication; however, GMD policy encourages use of persistent code identifiers (archive with doi) since a public github site could change in the future. Obtaining a doi for the already available github GC-YIB code should be relatively easy (e.g. via zenodo or other service that interfaces directly with github). A GC-YIB doi would be ideal.*

→ Thank you for your suggestions. We have generated doi link for GC-YIBs codes. In the Code Availability section, we share the link as follows:

"The source codes for the GC-YIBs model is archived at https://doi.org/10.5281/zenodo.3659346." (Lines 549-550)

In addition, we shorten the word 'version' in title to 'v':

Implementation of Yale Interactive terrestrial Biosphere model v1.0 into GEOS-Chem v12.0.0: a tool for biosphere-chemistry interactions